# Diabetes Control Status and Severity of Depression: Insights from NHANES 2005–2020

**DOI:** 10.3390/biomedicines12102276

**Published:** 2024-10-08

**Authors:** Raedeh Basiri, Yatisha Rajanala, Megan Kassem, Lawrence J. Cheskin, Cara L. Frankenfeld, Maryam S. Farvid

**Affiliations:** 1Department of Nutrition and Food Studies, George Mason University, Fairfax, VA 22030, USA; 2Institute for Biohealth Innovation, George Mason University, Fairfax, VA 22030, USA; 3Department of Health Administration and Policy, George Mason University, Fairfax, VA 22030, USA; 4Department of Global and Community Health, George Mason University, Fairfax, VA 22030, USA; 5Department of Medicine, Johns Hopkins School of Medicine, Baltimore, MD 21205, USA; 6Center for Interdisciplinary & Population Health Research, MaineHealth Institute for Research, Westbrook, ME 04092, USA

**Keywords:** diabetes, prediabetes, depression, HbA1c, NHANES, mental health, blood glucose control

## Abstract

**Background/Objectives:** Examining the risk of depression among patients with diabetes is crucial for understanding the mental health burden of this chronic condition. This study examined the likelihood of depression severity among participants in the National Health And Nutrition Examination Survey (NHANES) from 2005 to 2020, based on glycemic control status. **Methods:** Depression severity was categorized into three levels using the Patient Health Questionnaire-9 (PHQ-9), and glycemic control status was categorized into five groups based on prior diabetes diagnoses and hemoglobin A1c (HbA1c) levels. Using multinomial logistic regression models, the odds ratio (OR) and 95% confidence intervals (95%CIs) of various severities of depression by glycemic control status were calculated after comprehensive adjustments. **Results:** Out of 76,496 NHANES participants from 2005 to 2020, 37,037 individuals who met our inclusion criteria were analyzed. The likelihood of depression in individuals with prediabetes was not significantly different from those with normoglycemia. In contrast, participants with diabetes had a higher likelihood of having depression versus individuals with normoglycemia even when they kept their HbA1c within the normal range (lower than 5.7%). Among individuals with diabetes, those with HbA1c < 5.7% had a higher likelihood of mild depression (OR: 1.54, 95%CI: 1.02–2.34), while having HbA1c ≥ 10.0% was significantly associated with a greater likelihood of moderate to severe depression (OR: 1.53, 95%CI: 1.07–2.19) compared to those with HbA1c levels of 5.7–10.0%. **Conclusions:** Our findings highlight the need for a holistic approach to diabetes care that includes mental health considerations, especially for those who are at the extremes of the HbA1c spectrum.

## 1. Introduction

Diabetes and depression pose significant global health challenges, with diabetes (total of type-1 and type-2) affecting 1 in 11 adults worldwide, and depression having a lifetime prevalence of 11–15% [1]. Both conditions contribute to medical and economic burdens, with diabetes accounting for 12% of global health expenditure [2]. Diabetes often coexists with neuropsychiatric comorbidities, particularly depression [3]; however, it remains a neglected condition in patients with diabetes, directly impacting their quality of life. Depression ranks as the fourth leading cause of Disability-Adjusted Life Years (DALYs) in developed countries, while diabetes holds the eighth rank [4].

Depression, a mood disorder defined in the American Psychiatric Association’s *Diagnostic and Statistical Manual of Mental Disorders*, Fifth Edition (DSM-5), disrupts emotions, cognition, and behaviors [5,6]. Diagnostic criteria include the core symptom of diminished mood or anhedonia, plus at least four additional symptoms, namely feelings of guilt or worthlessness, fatigue, loss of concentration, suicidal thoughts, at least 5% change in weight, a change in psychomotor activity, or sleep disruptions lasting for at least two weeks [7].

The treatment of depression in patients with diabetes is challenging, often proving ineffective and sometimes refractory [8]. Further complicating matters, some antidepressant medications can adversely affect blood glucose control [9]. On the other hand, the demanding nature of diabetes management for patients, including the complexities of blood glucose control, medication adherence, and lifestyle adjustments, contributes to elevated stress levels. The emotional burden associated with navigating the potential complications of diabetes also significantly heightens a susceptibility to developing or worsening depression [10,11]. Coping with the uncertainty of managing a chronic condition and the potential complications associated with diabetes, such as neuropathy or cardiovascular issues, can further contribute to the development of depression [12].

Studies reveal an increased prevalence of depression in prediabetes and undiagnosed diabetes, with markedly higher rates in individuals with diagnosed diabetes compared to those with a normal glucose metabolism [13]. The interactions between the hypothalamus, pituitary gland, and adrenal glands can cause alterations to cortisol production, which could be an underlying mechanism of an increased depression risk in people with impaired glucose tolerance [14]. Chronic inflammation and fluctuations in blood glucose that could lead to mood swings and impact neurotransmitters could be other potential causes of depression in those with diabetes or prediabetes [15]. Inflammatory markers such as C-reactive protein (CRP) and interleukin-6 (IL-6) are often elevated in individuals with depression, diabetes, and prediabetes. This chronic low-grade inflammation can affect the brain and contribute to the development of depressive symptoms [16,17]. This increased risk is intricately linked to the complex interplay between physical health, lifestyle factors, and the psychological burden associated with managing a chronic condition like diabetes [18,19]. The bidirectional association between diabetes and depression suggests shared biological mechanisms, emphasizing the need for a comprehensive understanding to enhance treatment and outcomes for these interconnected conditions [20]. The early identification and comprehensive management of depression in individuals with diabetes is critical for improving the overall outcomes and enhancing the quality of life of this vulnerable population [21,22]. Thus, recognizing the risk factors of depression among patients with diabetes is essential, and addressing both conditions comprehensively can provide further benefits to overall health.

While the increased risk of depression among individuals with diabetes is widely reported, the association between potential contributing factors—including diabetes control status, anthropometric measures, and demographic variables—and varying degrees of depression severity remains largely underexplored. Thus, it is important to investigate the link between diabetes management and different levels of depression severity, while accounting for potential covariates and stratifying according to influential predictors. This comprehension can assist in identifying vulnerable populations and optimizing patient treatment approaches.

We aimed to assess the likelihood of experiencing varying degrees of depression across different categories of glycemic control status to identify patients with diabetes at the highest risk of depression after a comprehensive adjustment.

## 2. Materials and Methods

### 2.1. Source of Data

Data from the National Health And Nutrition Examination Survey (NHANES), a series of cross-sectional, nationally representative health examination surveys, spanning 2005 to 2020 were used [23]. The NHANES, conducted by the Centers for Disease Control and Prevention (CDC), assesses the health and nutritional status of the U.S. population through interviews and physical examinations. Each year, NHANES samples about 5000 individuals using a complex, multistage probability-sampling design to ensure national representativeness. The process begins with geographical stratification, dividing the country into Primary Sampling Units (PSUs) based on factors like region and urbanization. A random sample of PSUs is then selected, followed by smaller geographical segments within each PSU, and then households within these segments. Finally, individuals within these households are randomly chosen, with the oversampling of certain groups such as the elderly, African Americans, Asians, and Hispanics to ensure sufficient data for these populations. The survey is conducted annually, with different participants each year. However, some participants might be re-invited in subsequent survey cycles, although this is not common [24,25]. For this study, the data were obtained from 76,496 participants who had participated in NHANES from 2005 to 2020. Participants were eligible for our analysis if they were aged 18+ years, had available test results for hemoglobin A1c (HbA1c), had answered at least seven questions from Patient Health Questionnaire-9 (PHQ-9) [26], and were non-pregnant/non-lactating. The final analysis included 37,073 participants. Figure 1 details the inclusion and exclusion criteria.

Based on the participants’ responses to the DIQ010 question (“Other than during pregnancy, {have you/has SP}/{have you/has SP} ever been told by a doctor or health professional that {you have/{he/she/SP}has} diabetes or sugar diabetes?”) and the availability of data from LBXGH (glycohemoglobin (%)), they were placed in five glycemic control categories as follows: normoglycemia (no prior diagnosis, HbA1c < 5.7%), prediabetes (no prior diagnosis, HbA1c 5.7%– <6.5%), Diabetes (diagnosed for diabetes with HbA1c < 5.7%, Diabetes with 5.7% < HbA1c < 10.0%), and Diabetes with HbA1c ≥ 10.0% [27]. Details about the classification of participants into these categories are outlined in Figure 2.

The severity of depression was defined based on the participants’ PHQ-9 scores. The PHQ-9 is a screening tool for depression consisting of nine questions about symptoms experienced over the past two weeks. The total scores range from 0 to 27, with higher scores indicating more severe depression. Following NHANES instructions, the total score was determined based on the sum of scores from responses to questions DPQ010 to DPQ090 from the NHANES website [23]. If participants did not respond to more than two questions out of nine, then the data were considered invalid and excluded; however, if one or two questions remained unanswered, then the prorated score calculation method was utilized as suggested in the analytical notes of the PHQ-9 [26]:

Calculating prorated score for determining severity of depression:Partial raw score (PRS) = sum of available questionnaire scores(1)
Total score = (PRS × 9) / total number of answered questions(2)

The participants were then divided into three categories based on their scores from PHQ-9: no depression (score: 0–4), mild depression (score: 5–9), moderate and severe depression (score: 10–27) [26].

Information about age, sex, and race/ethnicity was obtained from the “Demographic Variables and Sample Weights” data, while the body mass index (BMI) was extracted from the “Body Measures” questionnaires. Participants were categorized into six BMI categories: underweight (<18.5 kg/m^2^), normal weight (≥18.5 and <25 kg/m^2^), overweight (≥25 and <30 kg/m^2^), class I obese (≥30 and <35 kg/m^2^), class II obese (≥35 and <40 kg/m^2^), and class III obese (≥40 kg/m^2^). Participants without information for their BMI were categorized as missing. Physical activity levels were evaluated based on the Metabolic Equivalent of Task (MET) hours per week (MET-hrs./week) using the Global Physical Activity Questionnaire (GPAQ). The physical activity questionnaire used by NHANES differed between the 2005–2006 period and 2007 onward. To ensure consistency in our analysis, we categorized the 2005–2006 population into four distinct groups based on their estimated physical activity intensity and aligned them with the MET categories applied to participants from 2007 onward. Further details on the categorization and alignment of physical activity data from 2005–2006 with the rest of the dataset are provided in the Appendix A. The MET-hrs./week was calculated for data from 2007 to 2020 based on the NHANES guidelines, which assign eight MET points for vigorous work and recreational activities and four MET points for moderate work, recreational activities, and walking/bicycling. The calculation of MET-hrs./week for physical activity involved using these MET scores, alongside the daily duration and frequency of the activities performed each week [28]. Participants with no physical activity (MET-hrs./week = 0) were placed in the lowest category, while the others were divided into four additional categories (quartiles), and a separate category was created for participants with missing data [28].

### 2.2. Statistical Analysis

The odds ratio (OR) and 95% confidence intervals (CIs) for the association between glycemic control status and severity of depression were evaluated by multinomial logistic regression models. The primary outcome was depression severity. Survey analytic techniques were used in descriptive and regression analyses to incorporate the complex stratified sampling procedure and population weighting [23]. All analyses were conducted using Stata software version 18, with statistical significance set at a two-sided *p*-value of <0.05.

#### 2.2.1. Examining the Association between Glycemic Control Status and Various Severities of Depression 

The odds of various severities of depression in individuals with prediabetes, diabetes with HbA1c < 5.7%), diabetes with HbA1c between 5.7% and 10.0%), and diabetes with HbA1c ≥ 10.0%) versus those with normoglycemia were evaluated. Model 1 was adjusted for age, sex, and race/ethnicity, while Model 2 included additional adjustments for BMI, physical activity, and the year of data collection alongside the covariates from Model 1.

#### 2.2.2. Examining the Associations between Diabetes Control Status and Various Severities of Depression 

The odds of various severities of depression related to diabetes control status were examined among participants with diabetes. This model was applied to assess the severity of depression in participants with diabetes across different HbA1c categories. Participants with HbA1c levels lower than 5.7%, as well as those with HbA1c ≥ 10.0%, were compared to participants whose HbA1c levels were between 5.7% and 10.0%.

#### 2.2.3. Examining the Association between Diabetes Control Status and Various Severities of Depression among Participants with Diabetes, Stratified by BMI and Race/Ethnicity 

We investigated whether the relationship between diabetes control status and depression differed by BMI (BMI < 25 kg/m^2^ vs. BMI ≥ 25 kg/m^2^) and race/ethnicity (Non-Hispanic White vs. other races/ethnicities).

## 3. Results

A total of 37,073 participants were eligible for inclusion in the study. The demographic distribution of the study population is reported in Table 1. In summary, 38.86% of the population with diabetes were 65 years or older. Non-Hispanic Whites were the largest ethnic group in the entire population (67.87%), as well as across all diabetes status subgroups. Among participants with diabetes, 52.19% were males. Overweight and obesity were prevalent among the general population of the study, with 32.50% being overweight and 37.15% in obesity classes I, II, or III.

### 3.1. Glycemic Control Status and Various Severities of Depression

In Model 1, the likelihood of depression in participants with prediabetes was significantly different than those with normoglycemia (OR for moderate to severe depression: 1.25, 95%CI: 1.09–1.45). However, this association didn’t remain significant when the data was further adjusted for BMI, year of data collection, and physical activity in Model 2. Additionally, a significantly increased likelihood of different severities of depression was observed in individuals with diabetes withHbA1c < 5.7%, 5.7% ≤ HbA1c < 10.0%, and HbA1c ≥ 10.0% compared to normoglycemia, in both Models 1 and 2. For instance, the odds ratio for moderate to severe depression was 2.00 (CI: 1.40–2.88), 2.14 (CI: 1.81–2.54), and 2.86 (95%CI: 2.07–3.94) for those with HbA1c < 5.7%, 5.7% ≤ HbA1c < 10.0%, and HbA1c ≥ 10.0%, respectively (Table 2, Model 1). This trend remained similar after adding more covariates in Model 2. A greater likelihood of mild depression was also observed in individuals with diabetes compared to the population with normoglycemia.

### 3.2. Diabetes Control Status and Various Severities of Depression

To examine the effect of diabetes control status on the likelihood of depression, participants with diabetes were compared across different categories of HbA1c levels. Compared to those with an HbA1c between 5.7% and 10.0%, the likelihood of mild depression (OR: 1.54, 95%CI: 1.02–2.34) was significantly higher among individuals with HbA1c < 5.7; while the likelihood of moderate to severe depression (OR: 1.53, 95%CI: 1.07–2.19) was significantly higher in those with HbA1c ≥ 10.0% compared to those with HbA1c levels between 5.7% and 10.0%, as shown in Table 3.

### 3.3. Diabetes Control Status and Various Severities of Depression among Participants with Diabetes, Stratified by BMI and Race/Ethnicity

Since BMI and race/ethnicity play an important role in diabetes control and depression [29,30], we also examined the associations between diabetes control status and severity of depression after stratification for these covariates. When stratifying the data by race/ethnicity (Non-Hispanic White vs. other racial/ethnic groups), significant positive associations between HbA1c < 5.7% and all severities of depression were observed only among the other racial/ethnic groups, while the results did not reach statistical significance in the Non-Hispanic White population. When stratified by BMI, significant positive associations were observed between HbA1c < 5.7% and mild depression in participants with a BMI ≥ 25 kg/m^2^, and moderate to severe depression in participants with a BMI < 25 kg/m^2^ (Table 4).

## 4. Discussion

Overall, having diabetes was associated with a higher likelihood of all severities of depression; however, no significant association was found between prediabetes and depression after adjustments for the year of data collection, BMI, and physical activity. In participants with diabetes and HbA1c < 5.7%, the likelihood of mild and moderate to severe depression was 1.90- and 1.70-fold higher than in those with normoglycemia, respectively. Similarly, among participants with HbA1c levels between 5.7% and 10.0%, the likelihood of mild and moderate to severe depression was 24% and 63%, higher, respectively, compared to those with normoglycemia. Additionally, for participants with HbA1c ≥ 10.0%, the likelihood of mild and moderate to severe depression was also notably high—1.49- and 2.31-fold greater than individuals with normoglycemia. Among participants with diabetes, those with HbA1c < 5.7% had a higher likelihood of mild depression (54%), while those with HbA1c ≥ 10.0% had significantly higher chance of moderate to severe depression (53%) than those with HbA1c between 5.7% to 10.0%. These results align with previous studies that reported the risk of depression as being twice as high among individuals with diabetes as among those without diabetes, as well as a higher risk of depression among patients with low and high levels of HbA1c [22,31]. Similarly, in our study population, the odds of depression were higher for individuals with HbA1c < 5.7% and HbA1c ≥ 10.0% compared to those with HbA1c between 5.7% and 10.0%.

We found that individuals with prediabetes had no statistically significant difference in the likelihood of depression compared to individuals with normoglycemia after adjustments for year of data collection, BMI, and physical activity. This differs from other studies, which found an increased risk of depression for the prediabetic population [32,33]. This disparity may be due to comprehensive adjustments in our study, differences in the duration of prediabetes, access to medical services, and other sociodemographic factors [34].

A 54% higher odds of mild depression and a 53% higher odds of moderate to severe depression observed in individuals with diabetes who had HbA1c levels below 5.7% or equal/above 10.0% compared to those with HbA1c levels between 5.7% and 10.0%, suggesting that both tighter glycemic control and higher blood glucose levels may be associated with an increased risk of depression. This may be due to the ongoing psychological stress of managing the condition, biological factors affecting mood regulation, social stigma, a reduced perception of control, and underlying psychological vulnerabilities [17,35]. These findings are aligned with other studies that suggest that the risk of depression in patients with diabetes is not linearly related to HbA1c levels [36,37]. Therefore, integrating mental health support into diabetes care, particularly for those who are at the extremes of the HbA1c spectrum, is crucial for addressing these complex interplays and improving overall well-being.

This data underscores the importance of a comprehensive approach to diabetes management that considers the mental health of patients with this challenging chronic disease in addition to their physical health. While maintaining lower HbA1c levels is recommended and beneficial for reducing the risk of diabetes-related complications, the challenges inherent in adhering to stringent dietary and medical restrictions may also be accompanied by an increased risk of depression and related complications. This aligns with the results of a study by Penckofer et al. [38], as it suggests that managing glycemic variability is not only important for physical health but also for mental well-being and overall quality of life. Healthcare providers should be aware of this potential risk and consider integrating regular mental health screenings and support into the care plan for patients with diabetes, including those who are achieving and maintaining lower HbA1c levels. This is important since it has been shown that depression may be a factor that highly influences the progression of diabetes complications [39].

These findings highlight the need for further research to understand the causal mechanisms behind these associations and to develop targeted interventions to help mitigate the risk of depression in patients with diabetes across the spectrum of glycemic control.

This study has multiple strengths, including the use of a nationally representative sample, the inclusion of 16 consecutive years of data (2005–2020), and comprehensive adjustments for potential confounding factors related to both diabetes and depression. Additionally, analyses were stratified by BMI and race/ethnicity to enhance the robustness of the findings.

The interpretation of the results should consider this study’s limitations. The approach we employed, which used cross-sectional data, limits our ability to determine the sequence and timing of events between diabetes and depression, thus restricting our ability to draw causal conclusions. Addressing the timing or two-way relationship between diabetes and depression is a significant gap that is crucial for understanding the underlying causes. Despite these methodological limitations, the present study sheds light on the intricate relationship between diabetes and depression. The findings suggest the need for tailored interventions addressing the varied needs of patient populations, encompassing both physical and mental health aspects. Long-term clinical trials are needed to identify optimal targets for HbA1c, BMI, and physical activity to effectively reduce the depression risk and enhance the overall well-being in this vulnerable group. Additionally, culturally sensitive interventions are essential to accommodate the diverse sociocultural backgrounds and experiences of patients suffering from diabetes, ensuring comprehensive and effective healthcare provision.

## 5. Conclusions

Our findings highlight the importance of diabetes control status as a risk factor for depression. Patients with HbA1c levels between 5.7% and 10.0% had the lowest odds for depression compared to all the other HbA1c groups. These results underscore the need for a holistic approach to diabetes care that includes mental health, particularly for those who are at the extremes of the HbA1c levels.

## Figures and Tables

**Figure 1 biomedicines-12-02276-f001:**
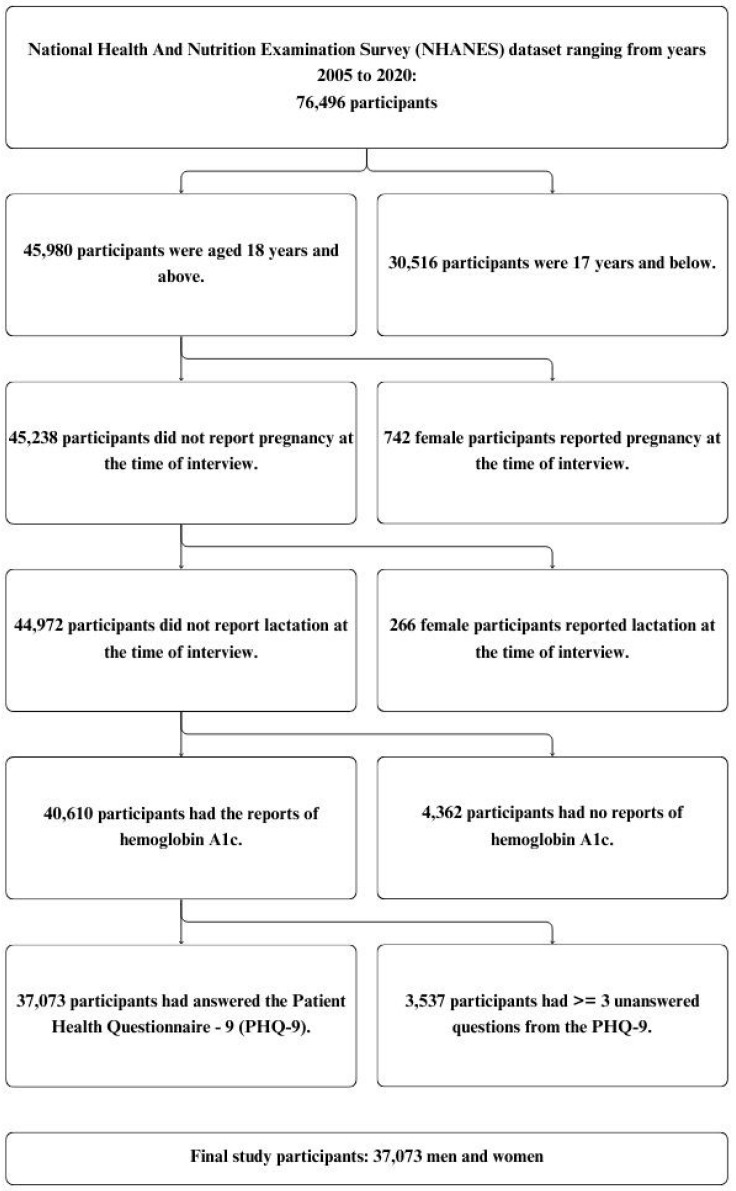
Flowchart of eligible individuals, the National Health And Nutrition Examination Survey (NHANES) 2005–2020.

**Figure 2 biomedicines-12-02276-f002:**
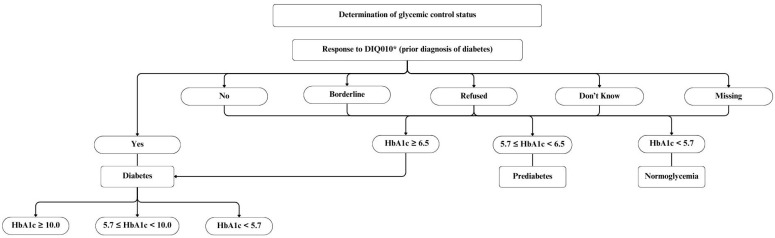
The determination of glycemic control status in the National Health And Nutrition Examination Survey (NHANES) 2005–2020. * DIQ010 question: “Other than during pregnancy, {have you/has SP}/{have you/has SP} ever been told by a doctor or health professional that {you have /{he/she/SP}has} diabetes or sugar diabetes?”.

**Table 1 biomedicines-12-02276-t001:** Characteristics of adult participants in the National Health And Nutrition Examination Survey (2005–2020) by glycemic control classification.

Variable (Number of Participants ^1^)	Total Participants (n = 37,073)	Normoglycemia (n = 22,393)	Prediabetes (n = 8815)	Diabetes (n = 5865)
Depression Categories				
No Depression	76.63%	77.80%	76.91%	69.47%
Mild Depression	15.53%	15.12%	15.01%	18.72%
Moderate to Severe Depression	7.84%	7.07%	8.08%	11.82%
Age Categories				
18–34 Years	28.38%	38.16%	9.86%	4.28%
35–44 Years	17.18%	19.55%	13.56%	9.82%
45–54 Years	19.02%	18.23%	20.88%	20.35%
55–64 Years	16.72%	12.72%	24.23%	26.70%
65+ Years	18.70%	11.34%	31.47%	38.86%
Sex				
Male	49.73%	49.90%	47.70%	52.19%
Female	50.27%	50.10%	52.30%	47.81%
Race/Ethnic Groups				
Non-Hispanic White	67.87%	70.73%	62.69%	60.39%
Non-Hispanic Black	10.68%	8.41%	15.59%	15.20%
Mexican American	8.45%	8.26%	8.29%	9.82%
Non-Mexican Hispanic	5.71%	5.70%	5.55%	6.08%
Other Races—Including Multi-Racial	7.29%	6.91%	7.88%	8.51%
Body Mass Index (BMI) Categories				
Normal Weight	1.60%	2.01%	0.99%	0.31%
Underweight	28.01%	33.96%	18.59%	10.14%
Overweight	32.50%	33.65%	33.08%	24.97%
Class I Obese	20.50%	18.14%	23.51%	28.83%
Class II Obese	9.54%	7.04%	13.17%	17.56%
Class III Obese	7.11%	4.57%	9.97%	16.75%
Missing	0.73%	0.62%	0.70%	1.44%
Physical Activity Categories				
No Activity	18.83%	14.65%	24.00%	33.80%
Quartile 1	19.25%	18.22%	21.56%	21.16%
Quartile 2	24.46%	25.85%	21.28%	21.93%
Quartile 3	18.60%	20.53%	15.95%	12.16%
Quartile 4	18.84%	20.72%	17.18%	10.95%
Missing	0.02%	0.03%	0.02%	0.00%

^1^ The participant numbers in the table are based on unweighted data, whereas the percentages are derived from weighted data.

**Table 2 biomedicines-12-02276-t002:** Odds ratio and 95% confidence intervals for the associations between glycemic control status and various severities of depression among participants in the National Health And Nutrition Examination Survey (n = 37,073).

Glycemic Control Status	Number	Depression Categories			
		Mild Depression		Moderate to Severe Depression	
		Odds Ratio	95%CI	Odds Ratio	95%CI
**Model 1 ***					
Normoglycemia	22,393	1 (reference)		1 (reference)	
Prediabetes	8815	1.07	0.96–1.19	1.25	1.09–1.45
Diabetes (HbA1C < 5.7%)	367	2.11	1.42–3.15	2.00	1.40–2.88
Diabetes (5.7% ≤ HbA1c < 10.0%)	4915	1.48	1.32–1.66	2.14	1.81–2.54
Diabetes (HbA1c ≥ 10.0%)	583	1.71	1.22–2.40	2.86	2.07–3.94
**Model 2 ****					
Normoglycemia	22,393	1 (reference)		1 (reference)	
Prediabetes	8815	0.97	0.87–1.09	1.10	0.95–1.28
Diabetes (HbA1C < 5.7%)	367	1.90	1.26–2.85	1.70	1.19–2.42
Diabetes (5.7% ≤ HbA1c < 10.0%)	4915	1.24	1.10–1.40	1.63	1.38–1.92
Diabetes (HbA1c ≥ 10.0%)	583	1.49	1.04–2.12	2.31	1.68–3.17

The participant numbers in the table are based on unweighted data. * Model 1 adjusted for sex (men and women), age (18–34, 35–44, 45–54, 55–64, and ≥65 years), race/ethnicity (Non-Hispanic White, Mexican American, Non-Mexican Hispanic, Non-Hispanic Black, and other races—including multi-racial). ** Model 2 adjusted for sex (men and women), age (18–34, 35–44, 45–54, 55–64, and ≥65 years), race/ethnicity (Non-Hispanic White, Mexican American, Non-Mexican Hispanic, Non-Hispanic Black, and other races—including multi-racial), BMI [underweight (<18.5 kg/m^2^), normal weight (≥18.5 and <25 kg/m^2^), overweight (≥25 and <30 kg/m^2^), class I obese (≥30 and <35 kg/m^2^), class II obese (≥35 and <40 kg/m^2^), class III obese (≥40 kg/m^2^), and missing values], year of collecting data (continuous), and physical activity (MET-hrs./week = 0, quartiles, missing).

**Table 3 biomedicines-12-02276-t003:** Odds ratio and 95% confidence intervals for the associations between diabetes control status and various severities of depression among participants with diabetes in the National Health And Nutrition Examination Survey (n = 5865).

Diabetes Control Status	Number	Depression Categories			
		Mild Depression		Moderate to Severe Depression	
		Odds Ratio	95%CI	Odds Ratio	95%CI
**Model 1 ***					
Diabetes (HbA1C < 5.7%)	367	1.46	0.99–2.16	0.94	0.65–1.36
Diabetes (5.7% ≤ HbA1c < 10.0%)	4915	1 (reference)		1 (reference)	
Diabetes (HbA1c ≥ 10.0%)	583	1.24	0.87–1.76	1.42	1.01–2.01
**Model 2 ****					
Diabetes (HbA1C < 5.7%)	367	1.54	1.02–2.34	1.07	0.75–1.54
Diabetes (5.7% ≤ HbA1c < 10.0%)	4915	1 (reference)		1 (reference)	
Diabetes (HbA1c ≥ 10.0%)	583	1.26	0.89–1.79	1.53	1.07–2.19

The participant numbers in the table are based on unweighted data. * Model 1 adjusted for sex (men and women), age (18–34, 35–44, 45–54, 55–64, and ≥65 years), race/ethnicity (Non-Hispanic White, Mexican American, Non-Mexican Hispanic, Non-Hispanic Black, and other races—including multi-racial). ** Model 2 adjusted for sex (men and women), age (18–34, 35–44, 45–54, 55–64, and ≥65 years), race/ethnicity (Non-Hispanic White, Mexican American, Non-Mexican Hispanic, Non-Hispanic Black and other races—including multi-racial), BMI [underweight (<18.5 kg/m^2^), normal weight (≥18.5 and <25 kg/m^2^), overweight (≥25 and <30 kg/m^2^), class I obese (≥30 and <35 kg/m^2^), class II obese (≥35 and <40 kg/m^2^), class III obese (≥40 kg/m^2^), and missing values], year of collecting data (continuous), and physical activity (MET-hrs./week = 0, quartiles, missing).

**Table 4 biomedicines-12-02276-t004:** Odds ratio and 95% confidence intervals for the associations between diabetes control status and various severities of depression among participants with diabetes in the National Health And Nutrition Examination Survey (n = 5865), stratified by race/ethnicity and BMI.

**Diabetes Control Status**	**Depression Categories**							
	**Mild Depression**				**Moderate to Severe Depression**			
	**Non-Hispanic Whites**		**Other Races/Ethnicities**		**Non-Hispanic Whites**		**Other Races/Ethnicities**	
	**Odds Ratio**	**95%CI**	**Odds Ratio**	**95%CI**	**Odds Ratio**	**95%CI**	**Odds Ratio**	**95%CI**
HbA1C < 5.7%	1.53	0.81–2.88	1.66	1.08–2.54	0.74	0.39–1.42	1.81	1.24–2.64
5.7% ≤ HbA1c < 10.0%	1 (reference)		1 (reference)		1 (reference)		1 (reference)	
HbA1c ≥ 10.0%	1.44	0.72–2.85	1.13	0.79–1.63	1.93	0.99–3.74	1.42	0.99–2.04
**Diabetes Control Status**	**Depression Categories**							
	**Mild Depression**				**Moderate to Severe Depression**			
	**BMI < 25 kg/m^2^**		**BMI ≥ 25 kg/m^2^**		**BMI < 25 kg/m^2^**		**BMI ≥ 25 kg/m^2^**	
	**Odds Ratio**	**95%CI**	**Odds Ratio**	**95%CI**	**Odds Ratio**	**95%CI**	**Odds Ratio**	**95%CI**
HbA1C < 5.7%	0.50	0.20–1.29	1.62	1.03–2.55	4.29	1.81–10.17	0.74	0.50–1.10
5.7% ≤ HbA1c < 10.0%	1 (reference)		1 (reference)		1 (reference)		1 (reference)	
HbA1c ≥ 10.0%	1.51	0.53–4.28	1.13	0.79–1.62	2.25	0.84–5.99	1.44	0.98–2.10

The models were adjusted for sex (men and women), age (18–34, 35–44, 45–54, 55–64, and ≥65 years), race/ethnicity (Non-Hispanic White and other ethnicities including Mexican American, Non-Mexican Hispanic, Non-Hispanic Black, and other races—including multi-racial), BMI [underweight (<18.5 kg/m^2^), normal weight (≥18.5 and <25 kg/m^2^), overweight (≥25 and <30 kg/m^2^), class I obese (≥30 and <35 and kg/m^2^), class II obese (≥35 and <40 kg/m^2^), class III obese (≥40 kg/m^2^), and missing values], year of collecting data (continuous), physical activity (MET-hrs./week = 0, quartiles, missing). Note: Race/ethnicity was not included as a covariate in the models when analyses were stratified by race/ethnicity, and similarly, BMI was excluded as a covariate in models stratified by BMI.

## Data Availability

Data are publicly available through CDC website; https://wwwn.cdc.gov/nchs/nhanes/ (accessed on 23 September 2024).

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
