# Peer review of "Diabetes Control Status and Severity of Depression: Insights from NHANES 2005–2020"

_biomedicines, 2024, doi:10.3390/biomedicines12102276_

Round 1

Reviewer 1 Report

Comments and Suggestions for Authors

The submission summarizes an analysis of several years of cross-sectional data on the association between diabetes status and depression. In its current form, the study conclusions are likely invalid and require substantial reconsideration of several analytic decisions taken during the data analysis phase. I will provide some specific comments below.

1.       The data are from NHANES which is a complex survey sample. The authors provide no information on how the sampling structure was considered in the analysis.

2.       The authors deviated from statistical best practice by categorizing both the exposure and the outcome measures. This is not a recommended strategy and is associated with known limitations. The authors should retain the data distributions to the extent practicable. For the exposure, three variables could be used which address the diagnosis status (2 dummy codes) and the AIC value. The outcome could simply be the observed score on the PHQ-9. This will have more statistical power and requires fewer assumptions than the categorization used in the current iteration. The authors could then generate conditional means of the outcomes for various combinations of diagnosis and A1C for interpretation.

3.       The authors report substantial missing data in the descriptive table but do not address how missing data are handled in the analysis.

4.       The authors interpret the results in the language of risk; however, since these data are cross-sectional, there is no temporal sequencing between diabetes and depressive symptoms, so it is not possible to determine which variable, if either, is increasing the risk of the other.

5.       Should the authors have a sufficient justification for the categorization of the depression outcome and maintain the multinomial logistic analysis, the OR estimates cannot be interpreted directly as increases in risk or prevalence since the OR estimates will overestimate the prevalence ratio. It would also be beneficial for interpretation to present the conditional probabilities of the outcome to facilitate the interpretation of effect size on an absolute and relative scale.

Author Response

Please see the response to the reviewer's comments in the attached Word document.

Thank you!

Reviewer 2 Report

Comments and Suggestions for Authors

Reviewer report_Basiri_Diabetes and Depression_biomedicines_2024

Thank you for the opportunity to review this manuscript. It presents an interesting study about the likelihood of experiencing varying degrees of depression across different categories of diabetes control status. 

The methodological approach is adequate. 

The introduction and discussion present a clarifying revision of the literature. However, I have some comments and suggestions presented below.

General comments

Please, review the whole text and avoid the use of bold font in the references. 

P value in the whole manuscript must be expressed as p value. 

Specific Comments

1. Introduction

Line 56. “Studies reveal an increased prevalence of depression in prediabetes and undiagnosed diabetes,”. Please, add some information in this issue. In the introduction you explain the relation between depression and diabetes considering the burden of the diabetes control influencing depression. But how the bibliography explains the relation between depression and prediabetes or undiagnosed diabetes? 

Line 78. “comprehensive”. Please, avoid the use of bold font. 

2. Materials and Methods

Line 81. Please add some information about the NHANES survey. For example, explain that it is performed in USA population. From all the states? 

Line 82: “spanning 2005 to 2020 was used [20]. The data was obtained from 85,750 participants who had taken the Patient Health Questionnaire (PHQ) [21].” From these data we have to suppose that in each of the year the survey was performed in different persons? Thus, there are non-repeated registers? Please, clarify. 

Figure 1. Please, include being “lactating mother” as an exclusion criterion also in the text, not only in the Figure 1. 

Figure 2. Please, is it possible to increase the size of the figure 2. It is not easily read. 

Line 106. “Total Raw Score (TRS) = (PRS * 9) / Total Number of Answered Questions” Please explain why do you make this operation: PRSx9, or is it the name of the questionnaire? If the PRS has 9 questions it must be explained previously.  

2.2. Model Construction and Statistical Analysis

Line 119. Model 2.2.1. How do you evaluated the risk of occurrence of various severities of depression in individuals with prediabetes and diabetes versus those without diabetes, if the three categories of HbA1c that you included in the model included diabetic people and non-diabetic and pre-diabetic were mixed: “HbA1c<6.5”

Please rewrite this section, explaining that you will made three models. It looks like because of the structure of the writing with two subsection that you will made only the last two. I propose to write a general paragraph explaining the software route and the test used, and after independently you explain each model. 

Moreover, explain the way of measuring physical activity. 

3. Results

Please, adapt the tables to the journal recommended format. 

4. Discussion

Line 229. Please syntax review: “Hence, future studies Hence, future studies are needed”.

Line 271. Please syntax review: “Long-term clinical trials to determine the best targets for HbA1c, BMI, and physical activity to reduce depression risk and improve overall well-being in this vulnerable group.”

Comments on the Quality of English Language

Minor editing of English language required

Author Response

Thank you for taking the time to review our manuscript. We have attached a Word document with our detailed responses to your comments.

Round 2

Reviewer 1 Report

Comments and Suggestions for Authors

The authors provided a conscientious revision to the manuscript. There only a couple remaining things to consider.

1. Were sampling weights used in the analysis? They should be used given the data source. If not, a strong justification should be presented in the method section.

2. The creation of a 'missing' category is acceptable for descriptive data presentation, but it is insufficient for a statistical analysis. The authors should consider a maximum likelihood or imputation-based approach. Alternatively, if data on physical activity are the largest source of missing data, it may be best to omit that data from the analysis. 

3. The justification for the continued reliance on categories is inadequate. I appreciate the authors providing output from a more appropriate model, but it makes sense to proceed with that model and then produce some model-implied estimates that would correspond to these meaningful categorizations. The use of cut points here ignores standard error of measurement and makes too many assumptions about within-group homogeneity and between-group heterogeneity to continue using them. 

Author Response

Thank you for taking the time to review our paper. We have carefully addressed all of your comments and revised the manuscript accordingly. Please find the attached document containing our detailed responses to your feedback.

We appreciate your thoughtful review.

Reviewer 2 Report

Comments and Suggestions for Authors

I want to thank the authors the correction of the manuscript. Now it is much better and I think it is ready for publication. Best regards

Author Response

Thank you for your kind words and for taking the time to review our manuscript. We truly appreciate your valuable feedback, which helped improve the paper. We are delighted to hear that you consider it ready for publication.

Round 3

Reviewer 1 Report

Comments and Suggestions for Authors

None